# Epidemiological Characteristics and Factors Associated with Alzheimer’s Disease and Mild Cognitive Impairment among the Elderly in Urban and Rural Areas of Hubei Province

**DOI:** 10.3390/jcm12010028

**Published:** 2022-12-20

**Authors:** Jing Cheng, Xiaoqi Ji, Lu He, Yutong Zhang, Tongtong Xiao, Qiang Geng, Zhihui Wang, Shige Qi, Fang Zhou, Jianbo Zhan

**Affiliations:** 1School of Public Health, Medical College, Wuhan University of Science and Technology, Wuhan 430065, China; 2Hubei Province Key Laboratory of Occupational Hazard Identification and Control, Wuhan University of Science and Technology, Wuhan 430065, China; 3Department of Epidemiology and Statistics, Chinese Center for Disease Control and Prevention, Beijing 102206, China; 4Hubei Provincial Center for Disease Control and Prevention, Wuhan 430079, China

**Keywords:** Alzheimer’s disease, mild cognitive impairment, epidemiology, protective factors, risk factors

## Abstract

Utilize the prevalence, associated factors and population distribution of AD and MCI among residents of the Hubei province aged 60 years or over to prove that elderly people who study and communicate with others, take part in regular physical exercise and choose a healthy lifestyle, will prevent or slow the decline in cognitive ability. If elderly people study and communicate with others, take part in regular physical exercise and choose a healthy lifestyle, can prevent or slow the decline in cognitive ability. A cross-sectional study was used for the recruitment of subjects. The screened patients with AD and MCI were then selected as patients in a case–control study. A total of 4314 subjects were recruited into the study. The prevalence of AD and MCI was 1.44% and 10.04%, respectively. The prevalence of AD and MCI differed significantly as a function of age and gender (*p* < 0.05). The preventative factors for AD and MCI, separately, included a happy marriage (OR = 0.69, 95%CI: 0.36–1.35) and higher education (OR = 0.65, 95%CI: 0.55–0.78). The risk factors for AD and MCI, separately, included infrequent participation in social activities (OR = 1.00, 95%CI: 0.60–1.66) and infrequent communication with children (OR = 1.35, 95%CI: 1.09–1.69). The prevalence of AD for people aged 60 or over in the Hubei province was lower than the national average of 3.06%. The prevalence of MCI was within the national range (5.2–23.4%). The influencing factors of AD and MCI were associated with the participants’ social connections, lifestyle behaviors, somatic diseases and so on. The elderly people who study and communicate with others, take part in regular physical exercise and choose a healthy lifestyle will prevent or slow the decline in cognitive ability. The conclusion section has been replaced.

## 1. Introduction

Alzheimer’s disease (AD) is an increasingly serious global health problem. In 1990, it was the 28th leading cause of death and by 2017, it had risen to the 8th leading cause of death [1]. It is estimated that more than 50 million people worldwide are currently living with dementia, and China has the largest number of dementia patients in the world. In 2020, there are about 176.03 million people aged over 65 years in China, accounting for 12.6% of the total population of 1.4 billion [2]. The population aging rate of China ranks first in the world [3]. It is estimated that by 2050, there will be over 30.3 million AD patients in China, 2.35 times the number of patients in 2015 [4]. The pathological changes of AD begin about 10 to 20 years before the onset of clinical symptoms [5], this is referred to as the mild cognitive impairment (MCI) stage. Those with MCI, especially those with memory problems and those with co-occurring MCI and anxiety, are more likely to develop Alzheimer’s disease (AD) or other dementias, as compared to those without MCI. Approximately 15 to 20% of people aged 65 years or older have MCI [6]. The prevalence of AD in people aged 60 or over in the Hubei province was lower than the national average of 3.06%. The prevalence of MCI was within the national range (5.2~23.4%).

AD is a serious threat to human health and quality of life due to its high morbidity, disability, and mortality rates, and brings a heavy burden on families and society [7]. To date, the pathogenesis of AD has not been determined, though the most widely recognized theories are the theory of amyloid protein (Aβ) toxicity and the theory of abnormal Tau protein [8,9,10]. Many studies have shown that the gradual loss of oligodendrocytes and myelin [11,12] is also a key factor leading to a decline in cognitive ability and pathological state repair ability. In current research, there is still no clear and effective way to prevent or treat AD. The current treatment method is to continuously stimulate patients and improve their psychological and mental states on the basis of conventional diet, sleep quality, psychological care, etc., combined with new methods such as memory recovery, language function recovery, and life function recovery, so as to promote the early recovery of patients.

A meta-analysis by Ward et al. [13,14] found that, on average, 32% of patients with MCI develop AD within five years, and about 10–20% of MCI patients progress to AD every year. Based on recent studies on the morbidity characteristics of dementia in related diseases (PSP, CBS), it can be found that MCI has a certain probability of conversion from PSP to CBS. However, some patients with MCI can become cognitively normal or stable without worsening through intervention and effective treatment [15,16], and about 24% of MCI patients are able to reverse to normal cognition (NC) [17].Therefore, it is very important to carry out early population-level screening, intervention, and possible treatment for MCI, so as to reduce the incidence and delay the occurrence of AD, thereby reducing the impact of AD on the health of the elderly population. Right now, we are looking at four treatments; for example, the pharmacological modulation of the glymphatic system, the therapeutic strategies (both natural and synthetic) targeting mitochondrial dysfunction, nutritional modifications to reduce the consumption of fruit, meat and processed products can be part of AD prevention and elucidation of interstitial water flow will be the key to developing therapeutic strategies for AD, especially with regard to prevention.

There are a large number of studies showing that having caregivers, age and residency are the main associated factors for the quality of life of patients with AD and MCI [17,18]. To identify factors associated with MCI and AD, the current study conducted a large-scale epidemiological field investigation of AD and MCI in the Hubei Province, which aims to provide a theoretical basis for the early intervention, prevention and control of AD and MCI.

## 2. Methods

### 2.1. Study Population

The population for this study was individuals aged 60 years or over who had a local household registration and had lived in the Hubei Province for more than one year. A multi-stage, stratified, clustered, random sampling method was used for recruitment. The specific sampling method includes the following five steps: Stage 1: Based on urban communities and rural township communities, Jingmen city and Huanggang City of Hubei Province were selected. Then, one county and one district were simply selected at random. Based on the sample size calculation formula: $n=\frac{Z^{2}p\left(1-p\right)}{\lambda^{2}}$ (*Z* is the statistic for significance test, *p* is the prevalence of AD in the elderly and *δ* is the tolerance error), *Z* was taken as 1.96 (α = 0.05), *δ* = *p* × 10% = 0.02, and *n* = 1536. To ensure sufficient sample size, an average of 2000 people were selected from each project county/district. Stage 2: One township and one street were randomly selected from each project county/district using PPS sampling method proportional to population size. Stage 3: A total of 4–8 administrative villages or neighborhood committees are selected by cluster sampling in the selected towns/streets. Stage 4: At least 100–200 households with members aged 60 or above in each administrative village (neighborhood committee) were randomly selected as the investigation households. Stage 5: Invite all permanent residents aged 60 or above to carry out the survey.

After multi-stage sampling, 2032 individuals from Qichun County, Huanggang City, Hubei Province were recruited, together with 2282 individuals from Jingmen City, Dongbao District, Hubei Province. After the preliminary screening of the survey participants, 58 patients with AD and 404 patients with MCI were identified and were selected as the case group.

### 2.2. Survey Methods and Contents

#### 2.2.1. Survey Methods

Potential AD and MCI subjects were identified through initial screening. Inclusion criteria to be included in this study were (1) age ≥ 60 years; (2) participants voluntarily agreed to cooperate with all investigators. Exclusion criteria were non-residents of the community; (3) those who refuse to provide correct information; (4) temporary residents; (5) patients with severe depression, schizophrenia and other mental diseases. Questionnaires, physical examinations, and laboratory tests were performed in a home-based and centralized manner by uniformly trained investigators. The Mini-Mental State Examination (MMSE) [19] and the Activities of Daily Living (ADL) [20] were used for the neuropsychological and daily activity ability assessment. There are 14 items on the ADL, with a scale of 0 to 100, with higher scores indicating better somatic functioning. The severity of the patient’s dementia is assessed by the MMSE, with a total score of 0 to 30, with lower scores indicating more severe dementia. The MCI diagnostic criteria are based on standards developed by the National Institute on Aging-Alzheimer’s Association (NIA-AA). The Geriatric Depression Scale (GDS) [21] and Hachinski Ischemic Score (HIS) [22] were used for the initial differential diagnosis of AD. The Clinical Dementia Rating (CDR) [23] was used to grade the severity of dementia. Finally, the suspected cases were further confirmed by neurology experts based on the preliminary diagnosis results combined with computed tomography (CT) evaluations. Additionally, then, a case–control study was performed. For the 58 diagnosed patients with AD and 404 diagnosed patients with MCI, gender and age (±2 years) were taken as matching factors. Elderly individuals with normal cognitive function who were matched to the cases with respect to area of residence were selected as the normal control (NC) group. The AD and NC groups were matched at a ratio of 1:4, with a total of 290 individuals across the AD and matched NC group, and the MCI group and NC group were matched at a ratio of 1:1, with a total of 808 individuals across the MCI and matched NC group.

#### 2.2.2. Survey Contents


(1)Inquiry and investigation. The main contents of the survey include: family status registration form and personal questionnaire survey. Among them, the contents of the family status registration form include family records, family demographic information (such as name, age, sex, education level, marital status, occupation, etc.), family elderly AD information, etc. Personal questionnaire contents include basic information such as demographic data, daily habits such as smoking, drinking, diet, physical activity and other behavior risk factors), primary hypertension-related information (such as common and treatment of chronic disease, chronic functional disorder, etc.), daily life ability and self-evaluation of health, and other major health problems such as dementia screening.(2)Physical examination. The examination content mainly includes: body and height measurement. Physical measurements include height, weight, waist circumference, blood pressure and heart rate. Height measurement using height sitting meter (maximum measuring length 2.0 m, accuracy 0.1 cm); electronic scales are used for weight measurement (maximum range 150 kg, accuracy 0.1 kg); waist circumference measurement using waist ruler (maximum measurement length 1.5 m, accuracy 0.1 cm); blood pressure and heart rate were measured using an electronic sphygmomanometer with an accuracy of 1 mmHg.(3)Laboratory testing. The detection content includes: routine blood biochemical index detection, namely fasting blood glucose and lipid. If the respondents can produce results of tests conducted within one year by other accredited medical institutions, these results are copied in the questionnaire. If not, fasting venous blood was collected from all subjects on site, and four tests of fasting blood glucose and blood lipid were performed.


#### 2.2.3. Conceptual Frameworks

A conceptual framework is constructed based on a literature review.



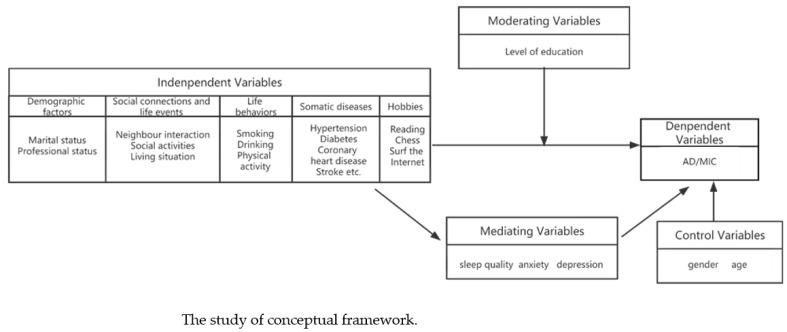



### 2.3. Data Analysis

Epidata 3.1 (version number 270108, Wuhan China) and SPSS 18.0 software were used for data processing and statistical analysis. Different descriptive statistics and hypothesis testing methods were used for the various types of statistical data. The rate and percentage were used to describe count data, and paired chi-square tests were used for between-groups comparisons. Measurement data were expressed as mean ± standard deviation (Mean ± SD). Multivariate conditional logistic regression models were used to analyze the association between the two groups and the various influencing factors. The potential risk factors were further screened by a step-by-step method, and the odds ratio (OR) and 95% confidence interval (CI) was calculated to evaluate the strength of the association between each factor and AD and MCI, respectively. All tests were two-tailed with α = 0.05; *p* < 0.05 was considered statistically significant.

### 2.4. Quality Control


(1)Before on-site investigation began, pre-investigation, revision, and improvement of a series of survey design schemes were performed.(2)During the process of the field investigation, there was a clear field workflow, a reasonable arrangement of on-site investigators, and timely feedback of the survey results.(3)After the completion of the field investigation, the quality of data entry was the focus. Each project site made local backups of the collected data every day and 5% of the questionnaires from each investigation site were extracted for review. Timely feedback was provided when problems were identified.


## 3. Results

### 3.1. Basic Information on the Sample

A total of 4314 people aged 60 years or over were recruited into this study from the Qichun County of Huanggang City and Dongbao District of Jingmen City, Hubei Province. A total of 304 invalid questionnaires were excluded from the analysis (including 81 people who refused to participate, 128 people lost to follow-up, 80 people who died, and 15 surveys with incomplete information). Thus, a total of 4010 respondents were included in the analysis, with a response rate of 92.95%.

Of the 4010 people included in the study, ranged in age from 60 to 110 years old, 2069 were female, and 1941 were male. The structure and category of the sample was shown in Appendix A.

### 3.2. The Prevalence of AD and MCI

#### 3.2.1. Overall Prevalence of AD and MCI and Average Score on MMSE

A total of 4010 people aged 60 years or over were included in this study, among which, 58 patients met the diagnostic criteria for AD, with an AD prevalence of 1.44%. Among the patients with AD, 17 were male (29.3%) and 41 were female (70.7%). A total of 404 patients met the diagnostic criteria for MCI, with an MCI prevalence of 10.07%. Among the patients diagnosed with MCI, 150 were male (37.1%) and 254 were female (62.9%). The average MMSE scores of the AD and MCI patients were 10.24 ± 0.59 and 18.75 ± 0.29, respectively. The MMSE scores were divided into six groups based on level of cognitive function. The specific scores are shown in Appendix A.

#### 3.2.2. Prevalence of AD and MCI as a Function of Age and Gender

With increasing age, the prevalence of AD and MCI first increased and then decreased; by ordinal logistic regression there was no statistically significant difference among the age groups (*p* > 0.05), which was shown in Table 1 and Table 2. 29.31% of AD patients are male and 70.69% are female, and 37.13% of MCI patients are male and 62.87% are female. (Figure 1 and Figure 2). The difference in the prevalence of AD/MCI between genders was statistically significant (*p* < 0.05).

#### 3.2.3. Prevalence of AD and MCI in Different Regions

Among the two areas selected for this study, Qichun County is a rural area and Dongbao District is an urban area. The prevalence of AD and MCI in the two areas was 1.44% and 10.04%, respectively. As shown in Appendix A, the proportion of AD among the elderly in rural areas was significantly higher than that in urban areas (96.55% vs. 3.45%; *p* < 0.05). There was no statistically significant difference in the proportion of MCI between the two groups (*p* > 0.05).

### 3.3. Factors Associated with the Presence of AD or MCI

#### 3.3.1. Demographic Factors Associated with the Presence of AD or MCI

Equilibrium analysis showed that there were no statistically significant differences in age and gender between the AD and MCI case groups and the respective control group (*p* > 0.05), indicating that the groups were comparable. The influence of other demographic factors on the prevalence of AD and MCI is shown in Table 3 and Table 4. The paired χ^2^ test showed that the incidence of AD differed significantly as a function of region and education level (*p* < 0.05). There were no significant differences in the prevalence of AD as a function of marital status and occupational status (*p* > 0.05). Univariate logistic regression analysis revealed that urban residence, happy marriage, higher education level, and previous relatively easy occupation were statistically significant, suggesting that these factors may be a protective factor for AD. Having a relatively easy occupation, higher education level, and happy marriage may also be protective factors for MCI; these relationships were statistically significant (*p* < 0.05).

#### 3.3.2. Influence of Social Connections and Life Events on the Incidence of AD and MCI

Univariate logistic regression analysis showed that a harmonious neighborhood relationship and living with family members were protective factors for AD and MCI. Frequent communication with children and continuing to work were also protective factors for AD. Active participation in social activities was a protective factor for MCI. However, active participation in social activities had no protective effect on AD. Frequent communication with children and ongoing work were risk factors for MCI. See Appendix A for details.

#### 3.3.3. Influence of Life Behavior on AD and MCI

The results of the univariate logistic regression analysis showed that a healthy lifestyle (including regular physical examination, wider waist circumference, frequent drinking of tea, and moderate alcohol) reduced the risk of AD, and frequent low to moderate physical exercise reduced the risk of MCI. Smoking and high levels of physical activity increased the risk of AD, while having sleep disturbances, regular smoking, drinking tea, and alcohol increased the risk of MCI. See Appendix A for details.

#### 3.3.4. Influence of Somatic Diseases on the Incidence of AD and MCI

Univariate logistic regression analysis showed that diabetes, stroke, disc herniation, hearing loss, and tumors were all protective factors for AD. High blood pressure, coronary heart disease, chronic bronchitis, asthma, tuberculosis, arthritis, and cervical spondylosis were risk factors for AD. Both coronary heart disease and stroke were risk factors for MCI. See Appendix A for details.

#### 3.3.5. Influence of Hobbies on AD and MCI

As can be seen from Appendix A, compared with the control group, there were no statistically significant differences in the rate of Internet surfing among the AD and MCI patients (*p* > 0.05), and there was no statistically significant difference in the rate of reading for interest between the controls and AD patients (*p* > 0.05). There was a significant difference in the rate of reading for interest between controls and the MCI case group (*p* < 0.05). There was also a statistically significant difference in the rate of playing chess and cards between controls and AD patients (*p* < 0.05), but no statistically significant difference between controls and MCI patients (*p* > 0.05). The results of the univariate logistic regression analysis showed that reading books and newspapers, playing chess and cards, and properly using computers to surf the Internet were all protective factors for AD and MCI.

#### 3.3.6. Multivariate Analysis of the Associated Factors AD and MCI

The single factors associated with AD and MCI as identified in the above analyses were included in a multivariate conditional logistic regression analysis. The stepwise method was used to enter variables. The results showed no statistically significant difference in the associated factors between the AD case group and the control group. Compared with the control group, higher education level, harmonious neighborhood relationship, and happy marriage were protective factors for MCI. Coronary heart disease, poor sleep quality, and frequent communication with children were risk factors for MCI. See Table 5 for details.

## 4. Discussion

This current study identified 58 patients with AD and 404 patients with MCI among the elderly population aged 60 years or above living in the Hubei Province. The overall prevalence of AD and MCI was 1.44% and 10.07%, respectively, which is significantly lower than the prevalence rates reported in other domestic populations [2,24,25]. The prevalence of AD in males was significantly lower than that in females (29.31% vs. 70.69%); the same was observed for MCI (37.13% vs. 62.87%). The prevalence of MCI in females was significantly higher than that of males for all five age groups in this study. This may be related to women’s longer life expectancy than men, lower estrogen levels after menopause, lower levels of education, less sleep (more housework) and more mood swings [26].

The possible factors associated with the occurrence of AD in this study included education, neighborhood associations, physical examinations, BMI, waist circumference, tea consumption, smoking, and playing chess and cards. Higher education level was a protective factor that reduced the risk of AD (OR = 0.07, 95%CI: 0.01–0.51). This is consistent with previous results [25]. During the process of education, blood flow to the brain will increase, which can effectively prevent nerve cell damage caused by free radicals [25]. Additionally, people with higher education, in contrast, tend to pay more attention to their physical health, making healthier lifestyle choices and reducing associated activities and lifestyle habits that may increase the risk of disease [27]. Education level also affects people’s ability and degree to participate in social behavior and social interaction. Frequent communication with neighbors was found to reduce the risk of AD. Elderly people who often communicate with neighbors may have a cheerful personality and strong affinity [28,29]. This may be because communicating with neighbors is a social activity that can promote the activity of the brain, enhance memory abilities [30]. Actively participating in physical examinations organized by community health service centers may also reduce the risk of AD. This may be because physical examination can provide early screening for hidden diseases, resulting in early detection and treatment of diseases among the elderly. In addition, BMI and waist circumference were negatively correlated with the severity of AD, and reasonable intake of nutrients may be a protective factor for AD [31]. Polyphenols in green tea have great potential for the prevention and treatment of neurodegenerative diseases in the elderly due to their strong antioxidant capacity. Feng et al. [32] conducted a cross-sectional study of the relationship between cognitive function and tea consumption in 716 elderly Chinese people aged 55 or above. The results showed that those who consumed more tea had better cognitive performance, and the protective effect of tea on cognitive function was not limited to specific types of tea. The current study indicated that smoking may increase the risk of AD. Studies have shown that nicotine and carbon oxide contained in tobacco can cause damage to the capillary endothelium, resulting in platelet agglutination and vascular stenosis, and increasing the risk of vascular obstruction, which can lead to dementia [33,34].

The factors affecting the occurrence of MCI in this study were education level, neighborhood communication, communication with children, coronary heart disease, sleep, and marital status. Higher education level was associated with a lower risk of MCI, making it a protective factor for MCI (OR = 0.67, 95%CI: 0.56–0.81). This is consistent with previous findings [10]. Frequent interaction with neighbors also reduced the risk of MCI (OR = 0.72, 95%CI: 0.59–0.89); this was also a possible protective factor for AD. In addition, a happy marriage was a protective factor for MCI (OR = 0.59, 95%CI: 0.40–0.88). This may be because in a happy marriage, the interaction and care with the spouse can prevent the development of MCI, while an unhappy marriage can lead to loneliness, depression and lack of love, which may promote the development of mild cognitive impairment. A recent study showed that maintaining physical and mental health helps to delay the decline in cognitive engagement in older adults experiencing cognitive impairment [35]. Sleep problems have been identified as a risk factor for cognitive decline and dementia. A recent meta-analysis identified ten sleep conditions or parameters associated with a higher risk of cognitive impairments [36]. Further, a recent study was found that participants who reported sleeping less than five hours had twice the risk of dementia as those who reported sleeping seven to eight hours per night [37]. Frequent communication with children in this study was associated with an increased risk of MCI (OR = 1.53, 95%CI: 1.19–1.97), which is contrary to the results of Ritchie [38]. This previous study argued that a lack of communication with one’s children leads to loneliness among the elderly, and can even lead to depression and improper diet, which will accelerate the occurrence and progress of MCI. The contrary result obtained in the current study may be related to the sample selected for this study. When children live with their parents, their communication increases, but supporting elderly parents will bring economic and mental pressure to children. Differences in living habits and values may lead to more conflicts, which will also increase the psychological burden of parents and may promote the occurrence of MCI.

Coronary heart disease was also found to be associated with MCI. Li et al. studied the relationship between cardiovascular disease and MCI and found that cardiovascular disease leads to cognitive impairment [39]. Thus, by actively controlling risk factors for vascular disease, we may be able to effectively prevent the occurrence of cardiovascular disease, and subsequently, the development of MCI.

Cognitive ability and self-care in Alzheimer’s patients have also been found to be associated with quality care [40]. Gallaway, P.J., et al. have also explored feasible methods to reduce the risk of mild cognitive impairment, Alzheimer’s disease and vascular dementia in the elderly [41], and Jia, R.X., et al. have also studied the influence of physical exercise on cognitive function in patients with Alzheimer’s disease [42].

Together with research on the pathogenesis, early diagnosis, and drug treatment of MCI and AD, research on factors influencing the incidence of MCI and AD, including diet and nutrition, may contribute to interventions for high-risk AD and MCI groups and improved prevention, early detection, early diagnosis, and treatment for AD and MCI. Such research also provides a scientific basis for the establishment of relevant government policies.

## 5. Strengths and Limitations

This is the first cross-sectional study to analyze the sporadic AD and MCI, as well as the influencing factors in urban and rural elderly people. Our study, conducted in Hubei Province, had a large sample size of the aged over 60 to explore epidemiological characteristics and influencing factors. Moreover, we obtained the prevalence data. The vast majority of AD cases have no known genetic cause, so it is crucial to identify environmental factors associated with the onset and progression of the disease. This study explores the living environment and lifestyle of patients with AD and MCI, combined with published studies to synthesize the risk factors and protective factors that are most likely to affect AD and MCI in older adults.

All of these provide valuable clues for follow-up research and preventive and intervening measures for AD and MCI.

This study also has several limitations (Table 6). First, 304 participants were lost to follow-up over the course of the study, which may have affected the results. Second, it is only a cross-sectional study, and the analysis results of risk factors have some limitations. For example, it is difficult to judge the time relationship and causality between the factors to be studied and the disease. Third, our sample results are not based on geographical distribution and economic level, the research results can only be representative of Hubei province. Fourth, our study only investigated the Han population, and whether the results can be extrapolated to ethnic minorities needs further study.

## 6. Conclusions

In this study, we found that the prevalence of AD in the elderly population aged 60 or over in the Hubei province was lower than the national average, which was significantly higher in females than in males. Additionally, the prevalence of MCI was within the range nationally. There were also differences in the prevalence of AD and MCI as a function of age, gender, and region. The influencing factors of AD and MCI were associated with their social connections, lifestyle behaviors and somatic diseases and so on. There is no effective treatment for AD and MCI, which poses a great challenge to the national and regional health, welfare, social services and other relevant departments, and has attracted the high attention of governments and health personnel. Therefore, how to generally carry out effective early screening, improve the early diagnosis rate and strengthen the standardized management of diseases, so as to delay the occurrence of diseases is the prevention and control strategy concerned by countries all over the world.

We believed that the elderly could be encourage to study, communicate, participate in social activities (such as reading books and newspapers, playing chess, writing, and other intellectual activities) and choose a healthy lifestyle and the government departments carry out capacity building training for medical and health workers in the disease control system and workers in primary medical and health service institutions, so as to prevent or slow the decline of cognitive ability.

## Figures and Tables

**Figure 1 jcm-12-00028-f001:**
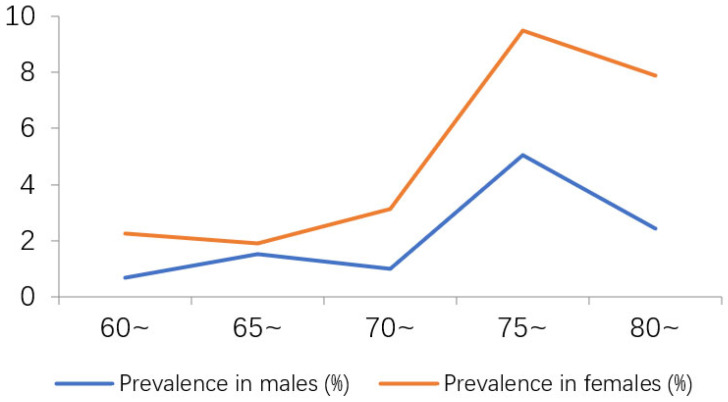
Trend in AD prevalence as a function of gender and age.

**Figure 2 jcm-12-00028-f002:**
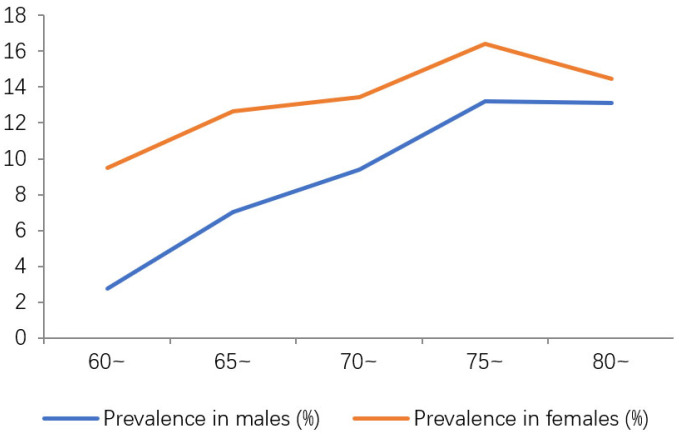
Trend in the prevalence of MCI as a function of gender and age.

**Table 1 jcm-12-00028-t001:** Prevalence of AD as a function of age and gender.

Age Group (Years)	Male		Female		Prevalence Rate (%)
	AD (*n*)	Prevalence Rate (%)	AD (*n*)	Prevalence Rate (%)	
60~	2	0.67	7	2.25	1.47
65~	4	1.54	5	1.92	1.73
70~	2	1.01	6	3.14	2.05
75~	6	5.04	13	9.49	7.42
80~	3	2.44	10	7.87	5.20
Total	17	1.70	41	3.99	2.86

**Table 2 jcm-12-00028-t002:** Prevalence of MCI as a function of age and gender.

Age Group (Years)	Male		Female		Prevalence Rate (%)
	AD (*n*)	Prevalence Rate (%)	AD (*n*)	Prevalence Rate (%)	
60~	14	2.73	65	9.49	6.60
65~	49	7.02	82	12.67	9.74
70~	33	9.40	47	13.43	11.41
75~	26	13.20	34	16.43	14.85
80~	24	13.11	26	14.44	13.77
Total	150	7.73	254	12.28	10.07

**Table 3 jcm-12-00028-t003:** Influence of different demographic factors on AD.

Variable	β	sx	χ^2^	P	OR (95% CI)
Region	−5.812	2.607	4.971	0.026	0.03 (0.00, 0.50)
Marital status	−0.367	0.340	1.167	0.280	0.69 (0.36, 1.35)
Level of education	−2.705	1.038	6.795	0.009	0.07 (0.01, 0.51)
Professional status	−3.461	2.744	1.592	0.207	0.03 (0.00, 6.79)

**Table 4 jcm-12-00028-t004:** Influence of different demographic factors on MCI.

Variable	β	sx	χ^2^	P	OR (95% CI)
Professional status	−0.934	0.252	13.788	0.001	0.39 (0.24, 0.64)
Level of education	−0.424	0.089	22.929	0.001	0.65 (0.55, 0.78)
Marital status	−0.452	0.183	6.117	0.013	0.64 (0.45, 0.91)

**Table 5 jcm-12-00028-t005:** Multivariate conditional logistic regression analysis of factors influencing MCI.

Variable	β	sx	χ^2^	P	OR (95% CI)
Coronary heart disease (CHD)	0.878	0.258	11.582	0.001	2.41 (1.45, 3.99)
Sleep	0.771	0.164	22.112	0.001	2.16 (1.57, 2.98)
Level of education	−0.399	0.096	17.183	0.001	0.67 (0.56, 0.81)
Neighbor interaction	−0.325	0.104	9.671	0.002	0.72 (0.59, 0.89)
Children communication	0.424	0.129	10.832	0.001	1.53 (1.19, 1.97)
Marital status	−0.529	0.203	6.777	0.009	0.59 (0.37, 0.88)

**Table 6 jcm-12-00028-t006:** The Strengths and Limitations.

Strengths	1. Had a large sample size
	2. Explores the living environment and lifestyle of patients with AD and MCI, combined with published studies to synthesize the risk factors and protective factors that are most likely to affect AD and MCI in older adults
Limitations	1. 304 participants were lost to follow-up over the course of the study, which may have affected the results
	2. It is only a cross-sectional study, and the analysis results of risk factors have some limitations
	3. Our sample results are not based on geographical distribution and economic level, the research results can only be representative of Hubei province.
	4. Our study only investigated Han population, and whether the results can be extrapolated to ethnic minorities needs further study

## Data Availability

The datasets used and/or analysed during the current study available from the corresponding author on reasonable request.

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
