# Peer review of "Epidemiological Characteristics and Factors Associated with Alzheimer’s Disease and Mild Cognitive Impairment among the Elderly in Urban and Rural Areas of Hubei Province"

_jcm, 2022, doi:10.3390/jcm12010028_

Round 1

Reviewer 1 Report

Authors of the study by Zhen et al elaborate on the epidemiological features of Alzheimer's Disease (AD) and Mild Cognitive Impairment (MCI) among elderly in the Hubei province. Authors indicate certain features which may impact the incidence e.g. sleep, level of education, neighborhood interaction, children communication and marital status. The issue is of interest, as the number of patients affected by AD and MCI gradually rises, however the work could be additionally further improved by:

1. Adding information on comorbidities found among the examined patients.

2. It would be valuable to stress that MCI may evolve to various types of dementia. In this context the clinical manifestation of MCI patients should be more specified, additionally authors could refer to recently discussed features impacting the incidence of dementia in AD and related diseases (PSP, CBS) as it is the case in the context of neuroinflammation. Perhaps enriching the discussion in this context would be valuable:

(A) ESR1 dysfunction triggers neuroinflammation as a critical upstream causative factor of the Alzheimer's disease process. Aging (Albany NY). 2022 Nov 1;14. doi: 10.18632/aging.204359. Epub ahead of print. PMID: 36326669.

(2) Neutrophil-to-lymphocyte ratio (NLR) at boundaries of Progressive Supranuclear Palsy Syndrome (PSPS) and Corticobasal Syndrome (CBS). Neurol Neurochir Pol. 2021;55(1):97-101. doi: 10.5603/PJNNS.a2020.0097. Epub 2020 Dec 14. PMID: 33315235.

3. The strenghts and limitations of the study could be presented in a table to make the global interpretation of the study more efficient. 

4. I believe that the conclusion section should be presented in a manner describing more future perspectives than recommendations.

5. The study should be verified by an English Native-Speaker.

Author Response

Dear editor,

I'm very sorry for replying to your email so late, because the system accidentally allocated your email to the dustbin, and I only saw it today. After seeing it, we immediately found it back and carefully revised the manuscript. Many thanks for your positive comments and valuable suggestions to improve the quality of our hardware. Please take time out of your busy schedule and give us another review. I hope you can give us a chance to learn and accept.

On behalf of my co-authors, we thank you very much for giving us an opportunity to revise our manuscript. We appreciate editor and reviewers very much for their positive and constructive comments and suggestions on our manuscript entitled “Epidemiological Characteristics and Factors Associated Alzheimer's Disease and Mild Cognitive Impairment among the Elderly in Urban and Rural Areas of Hubei Province”. We are very sorry to update the revised manuscript so late because the adding of some necessary experiment data.

Appended to this letter is our point-by-point response to the comments raised by the reviewers. The comments are reproduced and our responses are given directly afterward in a different color (red).

 We would like also to thank you for allowing us to resubmit a revised copy of the manuscript.

We hope that the revised manuscript is accepted for publication in the Journal of clinical medicine.

1.Adding information on comorbidities found among the examined patients.Thank you for the information suggested. Although relevant research has been done, I am sorry that I cannot provide specific information because it involves patient privacy.

2.It would be valuable to stress that MCI may evolve to various types of dementia. In this context the clinical manifestation of MCI patients should be more specified, additionally authors could refer to recently discussed features impacting the incidence of dementia in AD and related diseases (PSP, CBS) as it is the case in the context of neuroinflammation. Perhaps enriching the discussion in this context would be valuable.Thank you for the clinical manifestation of MCI patients suggested. The precedent version of the clinical manifestation of MCI patients has been added, adding information is based on recent studies on the morbidity characteristics of dementia in related diseases (PSP,CBS), it can be found that MCI has a certain probability of conversion from PSP to CBS.

3.The strenghts and limitations of the study could be presented in a table to make the global interpretation of the study more efficient. Thank you for the strenghts and limitations of the study suggested. The precedent version of strenghts and limitations of the study has been added, adding table

Strenghts

1.Had a large sample size

2.Explores the living environment and lifestyle of patients with AD and MCI, combined with published studies to synthesize the risk factors and protective factors that are most likely to affect AD and MCI in older adults

Limitations

1.304 participants were lost to follow-up over the course of the study, which may have affected the results

2.It is only a cross-sectional study, and the analysis results of risk factors have some limitations

3.Our sample results are not based on geographical distribution and economic level, the research results can only be representative of Hubei province.

4.Our study only investigated Han population, and whether the results can be extrapolated to ethnic minorities needs further study

  1. I believe that the conclusion section should be presented in a manner describing more future perspectives than recommendations.Thank you for the conclusion section suggested.The precedent version of the abstract conclusion section has been replaced, becomingthe elderly people to study and communicatewith others,take part in regular physical exercise and choose a healthy lifestyle,will prevent or slow the decline of cognitive ability.The conclusion section has been replaced, becoming believed.
  2. The study should be verified by an English Native-Speaker.Thank you for the study suggested.However,this article has been professionally polished.The attachment is as follows.And we hope the revised manuscript could be acceptable for you.

Thank you for your comments and suggestion concerning our manuscript. The comments and suggestions are all valuable and very helpful for revising and improving our paper, as well as the important guiding significance to our researches. We have studied comments carefully and have made correction which we hope meet with approval.

Sincerely,

CHENG Jing

Reviewer 2 Report

Please clearly state the purpose of the research.

The purpose of the research should be related to the title of the article.

Please provide your research questions and hypothesis.

What are the weaknesses of the research.

The conclusions are a repetition of the results.

They should be shortened.

What's new in research for treating Alzheimer's disease.

Author Response

Dear editor,

I'm very sorry for replying to your email so late, because the system accidentally allocated your email to the dustbin, and I only saw it today. After seeing it, we immediately found it back and carefully revised the manuscript. Many thanks for your positive comments and valuable suggestions to improve the quality of our hardware. Please take time out of your busy schedule and give us another review. I hope you can give us a chance to learn and accept.I'm looking forward to hearing from you.

On behalf of my co-authors, we thank you very much for giving us an opportunity to revise our manuscript. We appreciate editor and reviewers very much for their positive and constructive comments and suggestions on our manuscript entitled “Epidemiological Characteristics and Factors Associated Alzheimer's Disease and Mild Cognitive Impairment among the Elderly in Urban and Rural Areas of Hubei Province”. We are very sorry to update the revised manuscript so late because the adding of some necessary experiment data.

Appended to this letter is our point-by-point response to the comments raised by the reviewers. The comments are reproduced and our responses are given directly afterward in a different color (red).

We would like also to thank you for allowing us to resubmit a revised copy of the manuscript.

We hope that the revised manuscript is accepted for publication in the Journal of clinical medicine.

1.Please clearly state the purpose of the research.The purpose of the research should be related to the title of the article.Thank you for the purpose of the research suggested.The purpose has been replaced,the replacing information is utilising prevalence, associated factors and population distribution of AD and MCI among residents of Hubei province aged 60 years or over to prove the elderly people to study and communicate with others,take part in regular physical exercise and choose a healthy lifestyle,will prevent or slow the decline of cognitive ability.

  1. Please provide your research questions and hypothesis.Thank you for the questions and hypothesis suggested.We think is an excellent suggestion .We have re-written this part according to the Reviewer’s suggestion.The version of the question and hypothesis has been added, adding information is Hypothesis If the elderly people to study and communicate with others,take part in regular physical exercise and choose a healthy lifestyle,whether can prevent or slow the decline of cognitive ability.It is in line 18.
  2. What are the weaknesses of the research.Thank you for the research suggested.The weaknesses are mentioned in line 382 of the article.

4.The conclusions are a repetition of the results.Thank you for the results suggested.The purpose has been replaced,the replacing information is the elderly could be encourage to study ,communicate,participate in social activities (such as reading books and newspapers, playing chess, writing, and other intellectual activities) and choose a healthy lifestyle and the government departments carry out capacity building training for medical and health workers in the disease control system and workers in primary medical and health service institutions,so as to prevent or slow the decline of cognitive ability.

5.They should be shortened.Thank you suggested.The article conclusion is shorten.

6.What's new in research for treating Alzheimer's disease.

We sincerely appreciate the valuable comments.We checked the literature carefully and added more references on AD into the INTRODUCTION part in the revised manuscript.

1)pharmacological modulation of the glymphatic system

Research Evidence of the Role of the Glymphatic System and Its Potential Pharmacological Modulation in Neurodegenerative Diseases J. Clin. Med. 2022, 11(23), 6964

2)the therapeutic strategies (both natural and synthetic) targeting mitochondrial dysfunction.

Therapeutic Potential of Targeting Mitochondria for Alzheimer’s Disease Treatment J. Clin. Med. 2022, 11(22), 6742

3)Nutritional modifications to reduce the consumption of fruit, meat and processed products can be part of AD prevention.

The Concentration of Fibronectin and MMP-1 in Patients with Alzheimer’s Disease in Relation to the Selected Antioxidant Elements and Eating Habits J. Clin. Med. 2022, 11(21), 6360

4)elucidation of interstitial water flow will be the key to developing therapeutic strategies for AD, especially with regard to prevention.

Blood Cerebrospinal Fluid Barrier Function Disturbance Can Be Followed by Amyloid-β Accumulation J. Clin. Med. 2022, 11(20), 6118

Thank you for your comments and suggestion concerning our manuscript. The comments and suggestions are all valuable and very helpful for revising and improving our paper, as well as the important guiding significance to our researches. We have studied comments carefully and have made correction which we hope meet with approval.

Sincerely,

CHENG Jing

Reviewer 3 Report

This is very interesting study that explored protective and risk factors for MCI and AD for the Chinese population. Here, I would like authors to clarify the bullet points below before publication.

1.  In many places, authors wrote "65 AND order". It should be 65 OR older.

2. Two places were selected, one for representing rural and one urban area. The difference in terms of age, sex, education between two regions is expected and important to study prevalence of the AD. But I couldn't see tables showing those statistics.     

3. Authors reported no significant difference across age groups in prevalence of AD. Since age group is with ordinal scale, ordinal logistic regression is appropriate. Authors need to report analysis method adopted for the result.

4. Though authors mentioned the total number of participants in method, it would be helpful to note what was the base number that prevalence was calculated in Table 1 and 2.

5. Please check the number in the line 196. Is it 37.13% than 7.13?

6. Authors wrote about the effect of occupation in prevalence of AD/MCI. But there were no description how this variable was defined or surveyed. The result saying easy occupation is protective for AD/MCI sounds a bit contradictory with other studies. Please discuss this point as well in discussion such as types of occupations.

7. Line 245-250. Based on small number of AD patients (<50), the arguments for the relation between AD prevalence and that many of somatic diseases doesn't sound strong.

8. Authors mentioned the national average prevalence of AD/MCI in abstract. But those numbers were not mentioned in main text or referenced. Those should be included in main text as well.   

Author Response

Dear editor,

I'm very sorry for replying to your email so late, because the system accidentally allocated your email to the dustbin, and I only saw it today. After seeing it, we immediately found it back and carefully revised the manuscript. Many thanks for your positive comments and valuable suggestions to improve the quality of our hardware. Please take time out of your busy schedule and give us another review. I hope you can give us a chance to learn and accept.I'm looking forward to hearing from you.

On behalf of my co-authors, we thank you very much for giving us an opportunity to revise our manuscript. We appreciate editor and reviewers very much for their positive and constructive comments and suggestions on our manuscript entitled “Epidemiological Characteristics and Factors Associated Alzheimer's Disease and Mild Cognitive Impairment among the Elderly in Urban and Rural Areas of Hubei Province”. We are very sorry to update the revised manuscript so late because the adding of some necessary experiment data.

Appended to this letter is our point-by-point response to the comments raised by the reviewers. The comments are reproduced and our responses are given directly afterward in a different color (red).

We would like also to thank you for allowing us to resubmit a revised copy of the manuscript.

We hope that the revised manuscript is accepted for publication in the Journal of clinical medicine.

1.In many places, authors wrote "65 AND order". It should be 65 OR older.We are really sorry for our careless mistake.Thank you for your reminder.The precedent version of "65 AND order" has been replaced,becoming 65 OR older.

2.Two places were selected, one for representing rural and one urban area. The difference in terms of age, sex, education between two regions is expected and important to study prevalence of the AD. But I couldn't see tables showing those statistics. Thank you for the table suggested,but these data are represented in the table in the article.

3,Authors reported no significant difference across age groups in prevalence of AD. Since age group is with ordinal scale, ordinal logistic regression is appropriate. Authors need to report analysis method adopted for the result.Thank you for the analysis method suggested. We think is an excellent suggestion .We have re-written this part according to the Reviewer’s suggestion.The analysis method information is added,adding in line 212.

4.Though authors mentioned the total number of participants in method, it would be helpful to note what was the base number that prevalence was calculated in Table 1 and 2.We sincerely appreciate the valuable comments.However,the base number is mentioned in attachment.

5.Please check the number in the line 196. Is it 37.13% than 7.13?We feel sorry for our carelessness.In our resubmitted manuscript,the typo is revised.Thank for you correction.

6.Authors wrote about the effect of occupation in prevalence of AD/MCI. But there were no description how this variable was defined or surveyed. The result saying easy occupation is protective for AD/MCI sounds a bit contradictory with other studies. Please discuss this point as well in discussion such as types of occupations.We sincerely thank the reviewer for careful reading.Maybe because the sample size is a little small, there is a certain bias, which leads to a bit contradictory with other studies.

7.Line 245-250. Based on small number of AD patients (<50), the arguments for the relation between AD prevalence and that many of somatic diseases doesn't sound strong.We sincerely appreciate the reviewer’s suggestion .Maybe because the sample size is a little small, there is a certain bias.

8.Authors mentioned the national average prevalence of AD/MCI in abstract. But those numbers were not mentioned in main text or referenced. Those should be included in main text as well.It is a valid point that numbers were mentioned in main text or referenced.We are addedprevalence of AD/MCI is in line 54.

Thank you for your comments and suggestion concerning our manuscript. The comments and suggestions are all valuable and very helpful for revising and improving our paper, as well as the important guiding significance to our researches. We have studied comments carefully and have made correction which we hope meet with approval.

Sincerely,

CHENG Jing
